# Targeted Metabolomics Reveals Impact of N Application on Accumulation of Amino Acids, Flavonoids and Phytohormones in Tea Shoots under Soil Nutrition Deficiency Stress

Xuejiao Gong [1,2], Lanying Li [1,2], Lin Qin [3], Yingbo Huang [1,2], Yulong Ye [1], Min Wang [1], Yingchun Wang [1], Yaqiong Xu [1], Fan Luo [1,2,*] and Huiling Mei [4,*]

1  Tea Research Institute, Sichuan Academy of Agricultural Sciences, Chengdu 610066, China
2  National Agricultural Experimental Station for Soil Quality, Ya'an 625014, China
3  Institute of Quality Standard and Testing Technology Research, Sichuan Academy of Agricultural Sciences, Chengdu 610066, China
4  College of Resources and Environmental Sciences, Nanjing Agricultural University, Nanjing 210095, China
*  Correspondence: luofan4321@gmail.com (F.L.); mhling07@gmail.com (H.M.)

**Abstract:** The abundant amino acids and flavonoids in tea crucially contribute to its particular flavor and many health benefits. The biosynthesis of these compounds is significantly affected by carbon and nitrogen metabolism, which is regulated by the nitrogen conditions in the soil. However, exactly how N-starved tea plants use N absorbed from the soil for the biosynthesis of amino acids, flavonoids, and phytohormones is still little known. Here, tea plants that were deficient in nitrogen owing to long-term non-fertilization were subjected to a higher N application (300 kg/ha) or lower N application rate (150 kg/ha) as well as organic or inorganic N. The levels of 30 amino acids, 26 flavonoids, and 15 phytohormone compounds were analyzed using ultra-high-performance liquid chromatography quadrupole mass spectrometry (UPLC-Q-MS/MS). It was found that a continuous lack of fertilization generated a minimal availability of soil N; as a result, the yield and the theanine and soluble sugar contents were greatly decreased, while the accumulation of seven flavonoid compounds (e.g., epigallocatechin, vitexin, and genistein) increased notably. The levels of theanine, glutamate, and aspartate significantly increased with the supply of N, whereas multiple amino acids, such as alanine, phenylalanine, valine, etc., decreased, indicating that the absorption of nitrogen is preferentially used for the biosynthesis of theanine and glutamate-derived amino acids by a N-starved tea plant. Meanwhile, the changes in the accumulation of flavonoids in tea shoots with various N supplies clarified that a lower N application rate has a negative influence while higher N has a positive effect on the synthesis of flavonoids in a N-starved tea plant. In addition, following N supply, the N-deficient tea plant accumulated ABA (Abscisic acid), SA (Salicylic acid), JA (Jasmonic acid), CKs (Cytokinins), and ACC (1-Aminocyclopropanecarboxylic acid), at 2.03, 1.14, 1.97, 1.34, and 1.26 times, respectively, as high as those in a tea plant with normal fertilization. Furthermore, we performed the correlation network analysis among amino acids, flavonoids, and phytohormones. Its result confirmed that glutamate, aspartate, and hydroxyproline showed a significantly positive correlation with 8, 11, and 8 flavonoid compounds, respectively. Cis-OPDA (cis-12-oxo-phytodienoic acid) was also significantly negatively correlated with eight flavonoid compounds (e.g., naringenin, myricetin, and quercetin). Collectively, our tests suggested that a lower N application promotes the biosynthesis of the theanine and amino acids involved in theanine synthesis, thus inhibiting the accumulation of other amino acids, while greater N application promotes flavonoids in a N-starved tea plant.

**Keywords:** nitrogen deficiency; tea plant; amino acid; flavonoids; phytohormone

## 1. Introduction

The tea plant (*Camellia sinensis* L.) is an evergreen leafy bush with both good ecological function and high economic value and is widely cultivated in many countries and regions in the world. The amino acids and abundant flavonoids in tea leaves are the important material basis for the unique flavor and human health functions of tea [1]. The formation of these substances is closely related to the carbon and nitrogen metabolism in the tea plant [2]. Nitrogen is one of the "three elements" of nutrition, which largely determine the balance of carbon and nitrogen metabolism in the tea plant [3], thus affecting the accumulation pattern and regulatory mechanism of the flavor substances in tea leaves. Previous studies have found that sufficient N is more beneficial for enhancing nitrogen metabolism and promoting the accumulation of amino acids, while a N deficiency contributes to higher-level flavonoids [2,4,5]. Dong et al. [6] and Li et al. [7] concluded that a higher accumulation of flavonoid glycosides requires an appropriate nitrogen supply; nitrogen deficiency or excess nitrogen both resulted in a decreased content of flavonoid glycosides in tea shoots. In terms of N applications, the tea plant shows an evident predilection for ammonium [8], which promotes the accumulation of theanine, glutamate, and arginine, while the nitrate nitrogen promotes the expression of genes related to tea polyphenol and catechin synthesis, increasing the accumulation of flavonoids [9]. Sun et al. [10] showed that organic nitrogen significantly increased glutamine, quinic acid, and proline accumulation and drastically decreased the accumulation of organic acids, such as octadecanoic acid, hexadecanoic acid, and eicosanoic acid in tea shoots, compared to inorganic nitrogen, making for green tea with a higher sensory quality. Therefore, Xie et al. [11] suggested replacing 25% of chemical fertilizer N with organic fertilizer N. These findings provide us with a deeper understanding of the relationship between N nutrition and tea plant growth and tea quality, and also provide guidance for achieving high quality and high yield in tea plantations through fertilization in tea garden cultivation. However, the impact of N supply on the accumulation of amino acids and flavonoids in N-starved tea plants is still little known.

Plant hormones play an important role in plant growth and can be classified according to their main physiological functions into growth-promoting hormones, such as IAA (indole-3-acetic acid), CKs (cytokinins), GAs (Gibberellins), BRs (brassinosteroids), and stress-responsive hormones, such as SA (salicylic acid), ABA (abscisic acid), JA (jasmonic acid), and ACC (1-Aminocyclopropanecarboxylic acid) [12]. Numerous studies have shown that phytohormones not only regulate plant growth and development but are also involved in regulating the synthesis of secondary plant metabolites [13], such as ABA, JA, and SA, which can affect flavonoid metabolism by regulating the expression of structural genes [14,15] or transcription factor genes related to the flavonoid biosynthetic pathway [16]. In tea plants, Sun et al. [17] concluded that elevated levels of IAA, ACC, and ABA all inhibited catechin accumulation, but ABA and ACC up-regulated the expression of genes related to the phenylpropanoid pathway and flavonoid pathway, promoting the synthesis of non-catechin flavonoid substances, such as anthocyanins, while IAA negatively regulates these genes. Zhao et al. [18] were the first to report that phytohormones and miRNA can synergistically regulate tea-flavored substances; for instance, catechin, caffeine, and theanine contents were positively correlated with IAA, ABA, and JA, while they were negatively correlated with SA content. Changes in the plant growth environment can cause variations in the content of hormones in plants [19]. For example, Li et al. [20] confirmed that IAA, ABA, GA3, and BR may be involved in the response of tea plants to drought stress, while CsLYCE expression was significantly negatively correlated with ABA content under drought stress. Tea plants synthesized indole at > 450 ng/h during tea geometrid caterpillar attacks, thus promoting the synthesis of defensive secondary metabolites, such as jasmonic acid [21]. However, changes in the endogenous hormone content of tea plants under different nitrogen nutrient conditions, especially how the nitrogen supply affects endogenous hormones under nitrogen starvation conditions, have rarely been reported.

To address these issues, we conducted a split-plot experiment to reveal how N-starved tea plants use N absorbed from the soil for the biosynthesis of amino acids, flavonoids,

and phytohormones. In this study, we focused on the physiological responses of tea leaves to N deprivation and resupply by analyzing the changes in the accumulation of amino acids, flavonoids, and phytohormones in tea leaves, with the purpose of furthering our knowledge of the role of the N nutrient in tea plants.

## 2. Materials and Methods

### 2.1. Plant Materials and Growth Conditions

The test site is located at the National Agricultural Experimental Station for soil quality in Ya'an, Sichuan Province, China (N 30°16′, E 103°17′, altitude 766 m). It has a humid subtropical monsoon climate, with an average annual temperature of 15.8 °C, a frost-free period of 297 days, and an annual rainfall of about 1500 mm. The experimental tea plant in the study was about 12 years old, rather than using seedlings, and the variety was Mingshan 131, which is a commercial cultivar that is widely cultivated in Sichuan Province. The soil type in the tea garden was yellow loam. A split-zone trial was used with four consecutive years of no-fertilizer treatment (NF, trial area of about 0.75 hm$^2$) and normal fertilizer treatment (TF, trial tea plantation of about 0.15 hm$^2$, with N 300 kg/ha, P$_2$O$_5$ 54 kg/ha, and K$_2$O 72 kg/ha per year) from October 2016. The TF base fertilizer was applied in October–November and the follow-on fertilizer was applied in February–March, with the base fertilizer applied in the furrows (20 cm deep) and the follow-on fertilizer applied by spreading. On 23 October 2020, the soil samples were taken and then tested for physicochemical characteristics.

On 24 October 2020, when the tea plant was 16 years of age, five fertilizer treatments were conducted in the continuous non-fertilization zone: (1) no fertilizer treatment (NF); (2) high N treatment I (HN1) with the same fertilization level as TF; (3) high N fertilizer treatment II (HN2) with an equal total N application to that for HN1, inorganic N provided by a chemical fertilizer, and organic N provided by an organic fertilizer (these yielded an equal share of the total N); (4) low N treatment I (LN1) of N at 150 kg/ha, with N provided by a chemical fertilizer; (5) low N treatment II (LN2), with the same N application as LN1 and N being provided by an organic fertilizer. The date of the follow-on fertilizer application of the HN1, HN2, and LN1 treatments was 20 February 2021. The area of the tea plantation used for each treatment was about 400 m$^2$, and all were set up with 3 biological replications, randomly arranged between the different treatments, and spaced out by 2 tea plant rows. Management measures, such as harvesting and pruning, were the same for all treatments. The amount of N application in each treatment is shown in Table 1.

**Table 1.** N application in the different treatments.

| Treatment | Fertilizer and Dosage kg/ha | | N Supply kg/ha | | P$_2$O$_5$ Supply kg/ha | K$_2$O Supply kg/ha |
|---|---|---|---|---|---|---|
| | Base Fertilizer | After Manuring | Inorganic | Organic | | |
| HN1 | Chemical fertilizer 600 | Urea 360 | 300 | 0 | 54 | 72 |
| HN2 | Commercial organic fertilizer 7500 | Urea 325 | 150 | 150 | 75 | 75 |
| LN1 | Chemical fertilizer 600 | Urea 40 | 150 | 0 | 54 | 72 |
| LN2 | Commercial organic fertilizer 7500 | 0 | 0 | 150 | 75 | 75 |
| TF | Chemical fertilizer 600 | Urea 360 | 300 | 0 | 54 | 72 |
| NF | 0 | 0 | 0 | 0 | 0 | 0 |

Note: The N content of urea was 46.3%, while the N, P$_2$O$_5$ and K$_2$O contents of chemical fertilizers were 22%, 9%, and 12%, respectively, and the N, P$_2$O$_5$ and K$_2$O contents of the commercial organic fertilizers were 2%, 1%, and 1%, respectively.

On 18 March 2021, the first batch of one bud–two leaves spring growth was picked. After picking, one part of the tea leaves of each treatment was microwave-killed and roasted at 80 °C until fully dry, then stored at −4 °C in the laboratory for a soluble sugar content

assay, then one part was stored at −80 °C in liquid nitrogen for theanine detection and targeted metabolomics. Samples were numbered as HN1, HN2, LN1, LN2, NF, and TF according to treatment.

### 2.2. Standard Compounds and Other Chemicals

All standard compounds (purity ≥ 99 %) and formic acid were purchased from Sigma-Aldrich, Inc. (St. Louis, MO, USA). Acetonitrile and methanol were purchased from Merck KGaA (Darmstadt, Germany). All reagents were of HPLC grade. Ultrapure water was prepared using a Milli-Q system (Millipore, Billerica, MA, USA).

### 2.3. Sample Analysis

2.3.1. Measurement of Tea Garden Yield

On 18 March 2021, the first batch of one bud and two leaves was picked in the season of spring, weighed, and used to calculate the tea yield (kg/ha).

2.3.2. Determination of Soluble Sugar

The anthrone–sulfuric acid method was based on the method used by Zhang et al. [22]. First, 1 g (accurate to 0.0001) of the ground tea sample was weighed and added to 100 mL of boiling water, extracted in a boiling water bath for 30 min, filtered, and then washed repeatedly several times. The filtrate was combined to make 250 mL. Then, 2 mL of filtrate was taken and 8 mL of anthrone sulfate solution was added slowly, heated in a boiling-water bath for 7 min, and cooled rapidly, then the absorbance was measured at 620 nm.

2.3.3. Determination of Theanine Content

Theanine was detected using an ultra-high-performance liquid chromatography/triple quadrupole system (Waters ACQUITY UPLC I-Class/Xevo TQ-XS, Milford, MA, USA) according to the national standard of the People's Republic of China (GB/T 30987-2020). Deionized water with 0.01% formic acid and acetonitrile was used as the mobile phases A and B, respectively. An Acquity UPLC HSS T3 (2.1 mm × 100 mm, 1.8 μm, Waters, Milford, MA, USA) was used for chromatographic separation, with a column temperature of 40 °C and an injection volume of 2.0 μL, and a flow rate of 0.3 mL/min. The liquid phase gradient was as follows: 0–3 min, 5%–10% B; 3–9 min; 10%–13% B; 9–12 min, 13%–18% B; 12–13.5 min, 18%–50% B;13.5–14 min, 50%–90% B; 14–14.5 min, 90% B; 14.5–15 min, 90%–5% B; 15–17.5 min, maintained at 5% B.

The electrospray ionization (ESI) settings were: positive ion mode, capillary voltage 3.0 kV, source temperature 150 °C, flow rates of the cone gas and the desolvation gas, 150 and 1000 L/h, respectively.

2.3.4. Targeted Amino Acid Analysis by UPLC-TQ-MS

After grinding with liquid nitrogen, 60 mg of the sample was weighed and added to 0.5 mL of aqueous methanol acetonitrile (2:2:1, *v/v*), vortexed for 60 s, extracted twice (30 min each time) by ultrasonic extraction at low temperature, left for 1 h at –20 °C, and then filtered. The filtrate was then centrifuged for 20 min (14,000× *g* rcf, 4 °C) and the supernatant was stored at –80 °C for later measurement.

The tea extracts were analyzed using an ultra-high-performance liquid chromatography system (Agilent 1290 Infinity, Santa Clara, CA, USA) coupled to a mass spectrometer (AB SCIEX 5500 QTRAP, Concord, ON, Canada). The separation was performed using a Zic-HILIC column (2.1 mm × 150 mm, 3.5 μm, Merck, Darmstadt, Germany); the autosampler temperature was 4 °C, the column temperature was 40 °C, the flow rate was 250 μL/min, and the injection volume was 1 μL. The mobile phase A was 0.08% formic acid aqueous solution, with 25 mM ammonium formate, and B was acetonitrile with 0.1% (*v/v*) formic acid. The gradient elution was as follows: 0–12min, 90%–70% B; 12–18 min, 70%–50% B; 18–25 min, 50%–40% B; 30–30.1 min, 40%–90% B; 30.1–37 min, maintained at 90% B. The mass spectra were acquired using electrospray ionization in the positive

ionization modes. The ion source conditions were set as follows: source temperature 500 °C, ion source gas1 (Gas 1) 40, ion source gas2 (Gas 2) 40, curtain gas 30, IonSapary voltage floating 5500 V.A. The QC sample is set for every interval of a certain number of experimental samples, which is used to test and evaluate the stability and repeatability of the system. (see Supplementary Table S1 for details)

### 2.3.5. Targeted Flavonoids Analysis by UPLC-TQ-MS

First, 100 mg of the liquid nitrogen dried sample and 300 µL of a 70% methanol solution, as well as 10 µL of internal standard solution, were added to a homogenization tube for homogenate. Sonicate extraction was conducted at 4 °C for 30 min, then the mixture was centrifuged at 14,000× $g$ (10 °C for 20 min) to obtain the supernatant. Using a purification column (Ostro 96-well plate) to filter the supernatant under positive pressure, it was eluted once again with 200 µL of extraction solution, then the filtrate was frozen at −80 °C.

The tea extracts were analyzed using an ultra-high-performance liquid chromatography system (Waters I-Class LC, Milford, MA, USA) coupled to a mass spectrometer (AB SCIEX 5500 QTRAP, Concord, ON, Canada). Separation was performed using an ACQUITY UPLC BEH column (2.1 mm × 150 mm, 1.7 µm, Waters, Milford, MA, USA); the autosampler temperature was 4 °C, the flow rate was 400 µL/min, and the injection volume was 2 µL. The mobile phases A and B were deionized water with 0.1% formic acid and acetonitrile with 0.1% formic acid, respectively. The gradient elution was as follows: 0–3 min, 5%–20% B; 3–4.3 min, 20% B; 4.3–9 min, 20%–45% B; 9–11 min, 45%–98% B; 11–13 min, 98% B; 13–13.1 min, 98%–5% B; 13.1–15 min, 5% B. The mass spectra were acquired using electrospray ionization in positive and negative ion modes. The ion source conditions were set as follows: source temperature 500 °C, ion source gas1 (Gas 1) 55, ion source gas2 (Gas 2) 50, curtain gas 30, IonSapary voltage floating 5500 V (ESI+), −4500 V (ESI−). A QC sample is set for every interval of a certain number of experimental samples, which is then used to test and evaluate the stability and repeatability of the system. (see Supplementary Table S2 for details)

### 2.3.6. Targeted Phytohormones Analysis by UPLC-TQ-MS

The standards and internal standards were diluted with methanolic water solution in series concentrations to establish the standard curve. The sample was ground in liquid nitrogen and 100 mg of the sample was put into a 2 mL centrifuge tube. Then, 30 µL of the internal standard solution and 1170 µL of acetonitrile extract (containing 1% formic acid) were added, vortexed, and mixed thoroughly; the sample was sonicated for 25 min at a low temperature and protected from light, then left overnight at −20 °C and centrifuged at 14,000× $g$ for 20 min at 4 °C, then the supernatant was put into a purification column (Ostro 96-well plate). The sample was filtered under positive pressure in a 96-well plate, then eluted once by adding 200 µL of extraction solution to the wells, then the filtrate was blown dry under nitrogen at -80 °C. The sample was then purged with 200 uL of methanolic water (1:1, *v/v*), centrifuged for 20 min (4 °C, 14,000 $g$), and the supernatants were collected for mass spectrometry analyses.

The tea extracts were analyzed with an ultra-high-performance liquid chromatography system (Waters I-Class LC, Milford, MA, USA) coupled to a mass spectrometer (AB SCIEX 5500 QTRAP, Concord, ON, Canada). The separation was performed using an ACQUITY UPLC BEH column (2.1 mm × 150 mm, 1.7 µm, Waters, Milford, MA, USA); the autosampler temperature was 4 °C, the column temperature was 45 °C, the flow rate was 400 µL/min, and the injection volume was 2 µL. Deionized water with 0.05% formic acid and acetonitrile with 0.05% formic acid were used as mobile phases A and B, respectively. The gradient elution was as follows: 0–10 min, 2%–98% B; 10–10.1 min, 98%–2% B; 11.1–13 min, maintained at 2% B. The mass spectra were acquired using electrospray ionization in positive and negative ion modes. The ion source conditions were set as follows: source temperature 500 °C, ion source gas1 (Gas 1) 45, ion source gas2 (Gas 2) 45, curtain gas 30, IonSapary voltage floating 5500 V (ESI+), −4500 V (ESI−). A QC sample was set for every interval of a certain number of experimental

samples, which was used to test and evaluate the stability and repeatability of the system. (see Supplementary Table S3 for details).

### 2.4. Data Processing and Statistical Analysis

The chromatographic peak areas and retention times were extracted using the Analyst TF 1.17 software (AB SCIEX, Concord, ON, Canada); retention times were corrected using the standards for metabolite identification, and contents were calculated from the standard curve.

The SPSS 26.0 software (IBM, Chicago, IL, USA) was used for LSD (least significant difference) analysis as well as a two-tailed Student's *t*-test [23]. Correlation analysis was performed using the Hmisc package in the R package software (version 4.1.2), then we visualized the correlation network using the Gephi software (version 0.9.2).

## 3. Results and Analysis

### 3.1. Yields of Tea with Different N Supply

Tea garden yield can be used to measure the nutrient abundance or deficiencies in a tea garden. As shown in Figure 1, fertilization has a significant effect on increasing yield, which increased with increased nitrogen application by 30.93%–134.23%, compared to NF. However, the yield of HN1 was significantly lower than that of TF, indicating that a long-term lack of fertilization caused nutrient stress in the tea plants. Among the different fertilization treatments, the total amount of nitrogen applied was 300 kg/ha, and the yield of the tea garden with inorganic nitrogen application (HN1) was significantly higher than that with an organic–inorganic nitrogen-mixed application (HN2), indicating a higher utilization efficiency of inorganic nitrogen by tea plants than that of organic nitrogen. Moreover, there was no significant difference in yield between the organic nitrogen treatment (LN2) and inorganic nitrogen treatment (LN1) when the total amount of nitrogen was 150 kg/ha.

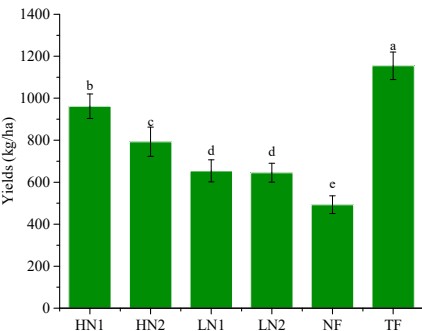

**Figure 1.** Yields with different N supplies. The statistical approach used was LSD (least significant difference) analysis. Every group contains 3 tea leaf samples. Different lower-case letters above the bars indicate significant differences among treatments (*p* < 0.05).

### 3.2. The Contents of Soluble Sugar and Theanine in Tea Shoots with Different N Supplies

It can be seen from Figure 2 that the contents of theanine and soluble sugar in tea leaves treated with NF significantly lower than that with TF, by 60.17% and 18.82%, respectively, indicating that long-term non-fertilization weakened the carbon assimilation capacity and nitrogen metabolism in tea plants. With a nitrogen supply, the contents of theanine and soluble sugar increased, reaching higher levels in HN1 and LN1, respectively. The content of theanine in HN1 and content of soluble sugar in LN1 increased by 48.64% and 11.52%, respectively, compared to NF.

### 3.3. Accumulation of Amino Acids in Tea Shoots with Different N Applications

Analyses of the differences in the accumulation of 28 kinds of amino acid components in tea shoots under different nitrogen supply conditions are shown in Figure 3. Compared to TF, the expression levels of alanine, phenylalanine, citrulline, creatine, and hydroxyproline

in non-fertilized tea gardens (NF) were significantly increased, while the expressions of glutamate and aspartate were significantly decreased. Compared to no fertilization (NF), the application of inorganic nitrogen at 300 kg/ha and 150 kg/ha (named HN1 and LN1) was up-regulated in terms of the expressions of aspartate, glutamate, glutamine, and putrescine, while a total of 16 amino acids, alanine, phenylalanine, tyrosine, aminoadipic acid, arginine, hydroxyproline, proline, leucine, isoleucine, valine, asparagine, citrulline, glycine, lysine, creatine, and ornithine, were down-regulated. In contrast, with the organic–inorganic mixed application of N 300 kg/ha (HN2), compared to HN1 and LN1, most of the up-regulated amino acids in the above-mentioned HN1 and LN1 were down-regulated, and most of the down-regulated amino acids were up-regulated. It was shown that HN2 has less effect on amino acid accumulation patterns in N-starved tea plants. On the other hand, all nitrogen fertilization treatments in long-term unfertilized tea gardens promoted the accumulation of aspartate, glutamate, or glutamine, indicating that nitrogen fertilization has activated the uptake and utilization of nitrogen in soil by tea plants.

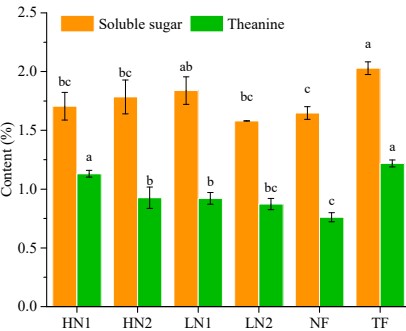

**Figure 2.** The contents of soluble sugar and theanine in tea shoots. The statistical approach used was LSD analysis. Every group contains 3 tea leaf samples. Different lower-case letters above bars indicate significant differences among treatments ($p < 0.05$).

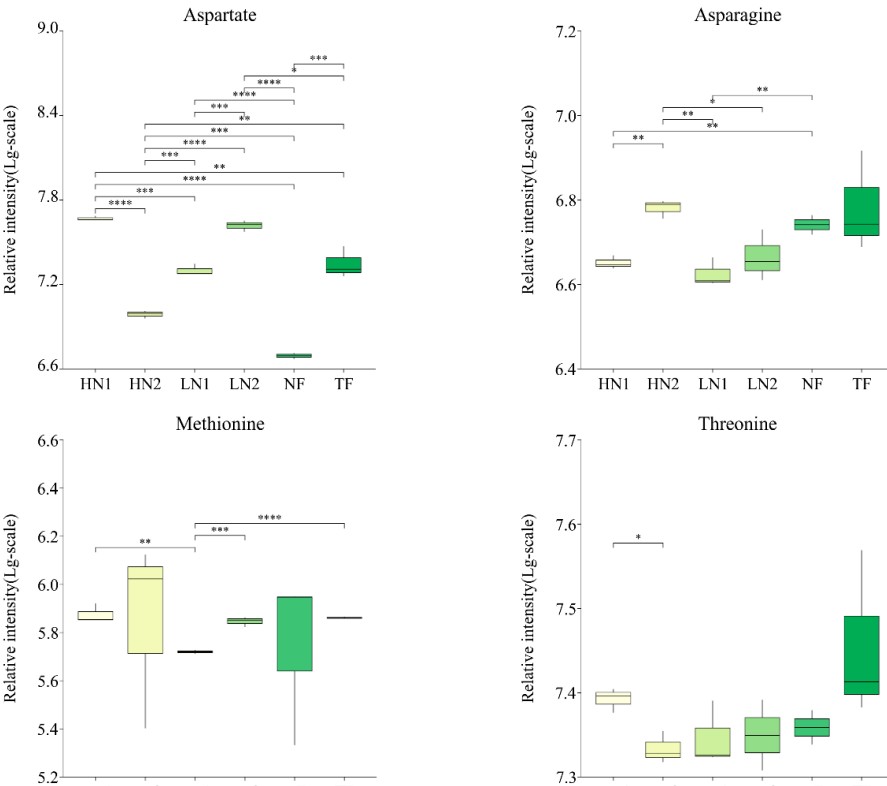

**Figure 3.** *Cont.*

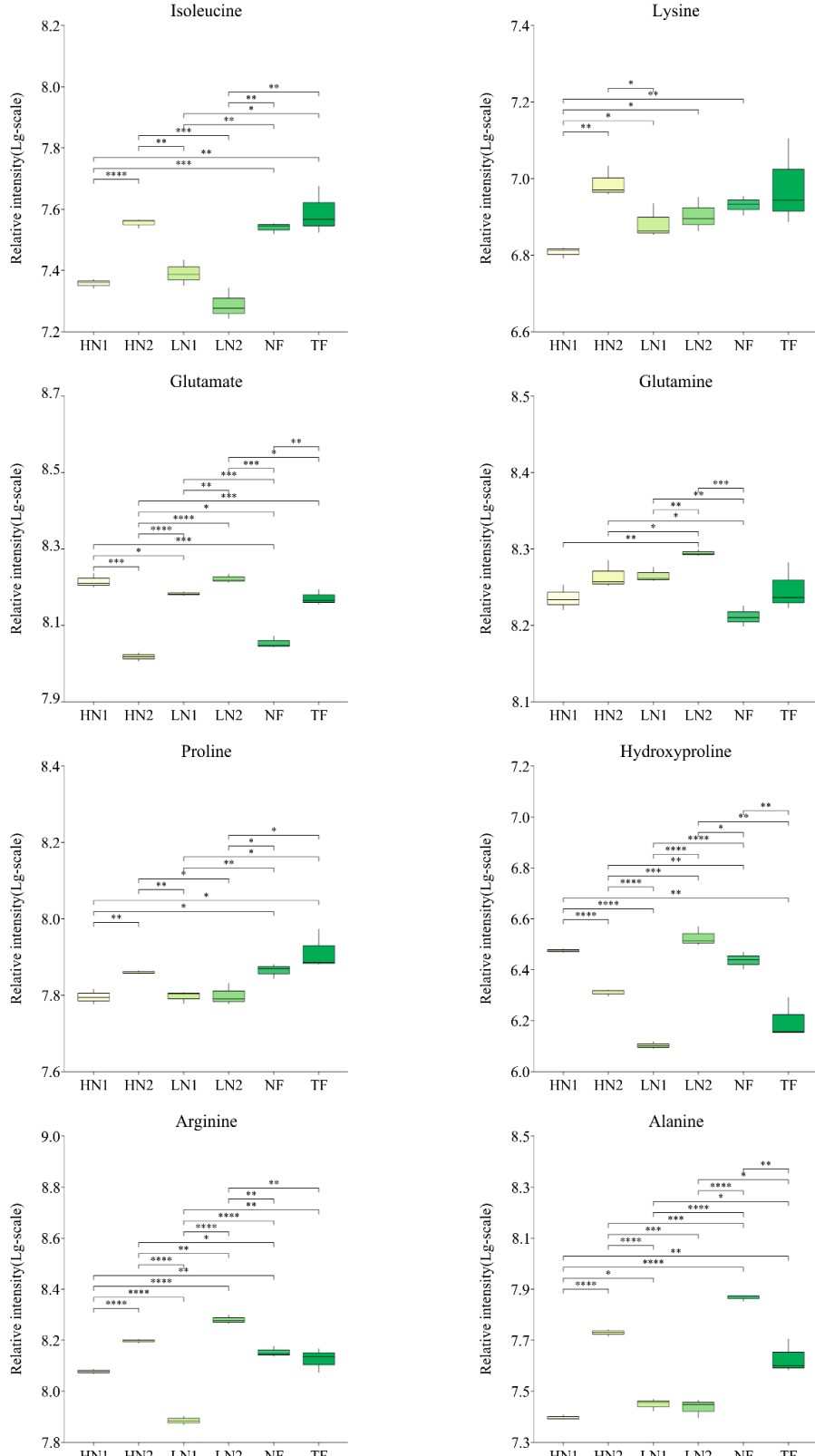

**Figure 3.** *Cont*.

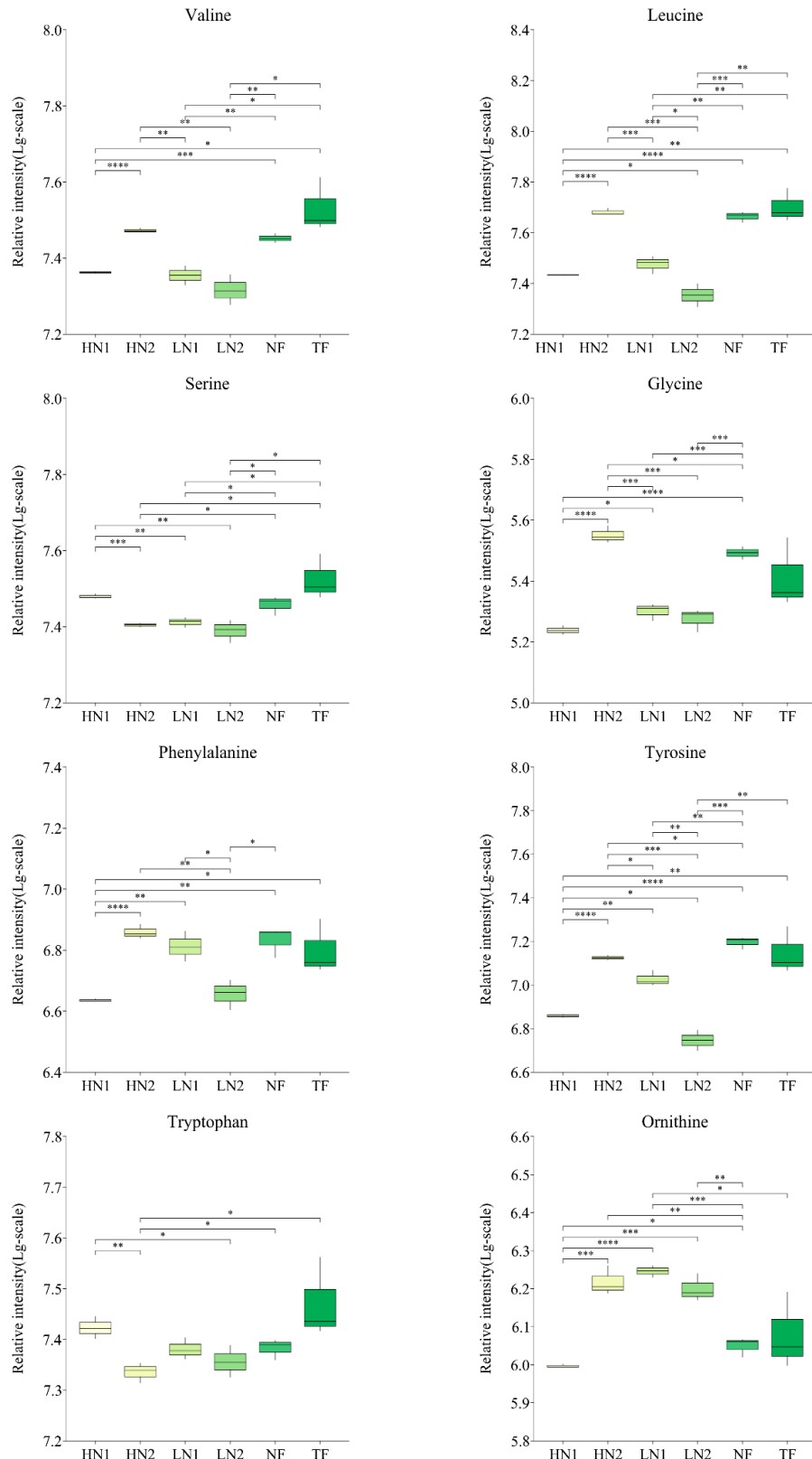

**Figure 3.** *Cont.*

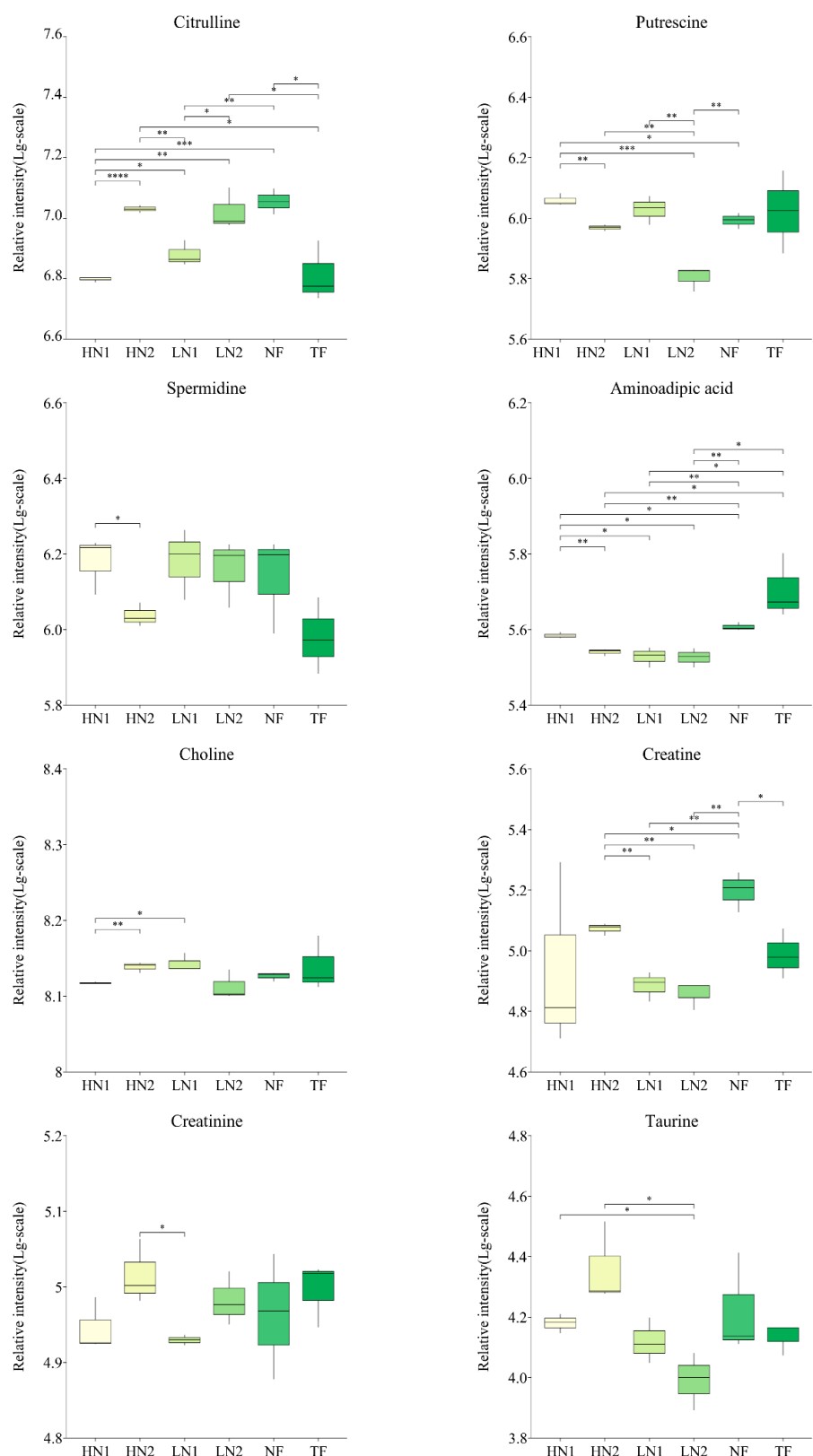

**Figure 3.** Changes of amino acids accumulation in tea leaves with different N supply. In total, 30 kinds of amino acid components were detected, and this figure shows only those with significant differences. The statistical approach used was a two-tailed *t*-test. Every group contains 3 tea leaf samples. ****, ***, **, * indicates a significant difference at *p* < 0.0001, *p* < 0.001, *p* < 0.01, and *p* < 0.05, respectively.

Moreover, the expression levels of hydroxyproline, aspartate, and spermidine in HN1 were significantly higher than those in TF, while the expression levels of alanine, leucine, isoleucine, valine, phenylalanine, proline, and tyrosine were significantly lower than those in TF, indicating a difference in response to the nitrogen nutrient of amino acid metabolism between a N-starved tea plant and a N-sufficient tea plant.

Spermidine and its precursor, putrescine, are important polyamines in plant responses to salt stress. In HN1 treatment, the expression of putrescine increased by 16.65%, compared to NF, and spermidine expression increased by 56.99%, compared to TF. However, the spermidine expression of HN2 had no significant differences compared to NF and TF, decreasing by 28.59% compared to HN1. This suggested that the application of high amounts of inorganic nitrogen caused salt stress.

### 3.4. Accumulation of Flavonoids in Tea Shoots with Different N Applications

As shown in Figure 4, the abundance of the 26 flavonoids detected showed significant differences among the different nitrogen fertilization treatments. Non-fertilized (NF) leaves promoted the accumulation of epigallocatechin (EGC), vitexin, apigenin, chrysin, genistein, genistin, and naringenin, while they inhibited the accumulation of sakuranetin, eriodictyol, naringin, and dihydrokaempferol. When organic nitrogen or inorganic nitrogen were applied at 150 kg/ha, LN1 and LN2 showed significant differences in 11 and 5 flavonoids compared to NF, of which 9 and 4 flavonoids were downregulated. Meanwhile, the abundance of 12 kinds of flavonoid components was significantly different in LN1 than in LN2, all of which were downregulated. It was indicated that nitrogen absorption inhibited the synthesis of flavonoids in N-starved tea plants with a small amount of N application. Compared to the application of organic–inorganic-mixed N 300 kg/ha (HN2) and inorganic N 150 kg/ha (LN1), there were significant differences in the cumulative amounts of 16 and 14 flavonoid components in the application of inorganic N 300 kg/ha (HN1), respectively, among which 14 and 12 were up-regulated. The accumulation of 9 flavonoid components (catechin, epicatechin, gallocatechin, dihydrokaempferol, rutin, myricetin, taxifolin, naringin, and isoliquiritigenin) in HN1 was also significantly higher than that in NF, indicating that the high inorganic N supply promoted the synthesis of flavonoids in N-starved tea plants. It is worth noting that the total amount of N applied in HN2 was equal to that in HN1, but the accumulation of 10 flavonoids in HN2 was downregulated compared to NF, showing closer effects on flavonoids with LN1. These analyses indicate that the amount of available nitrogen was the main factor causing differences in the accumulation of flavonoids in this study.

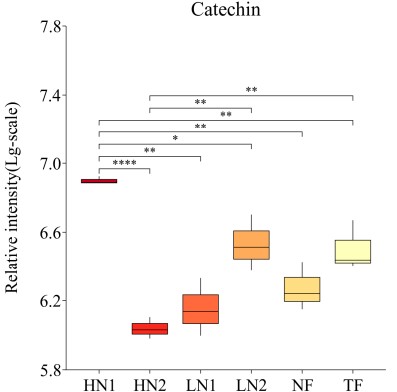 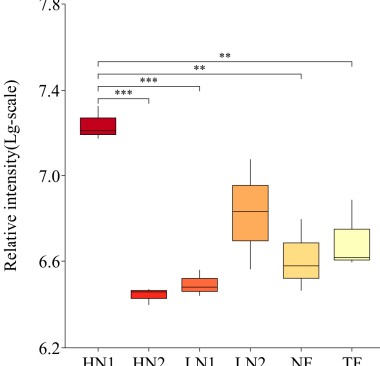

**Figure 4.** *Cont.*

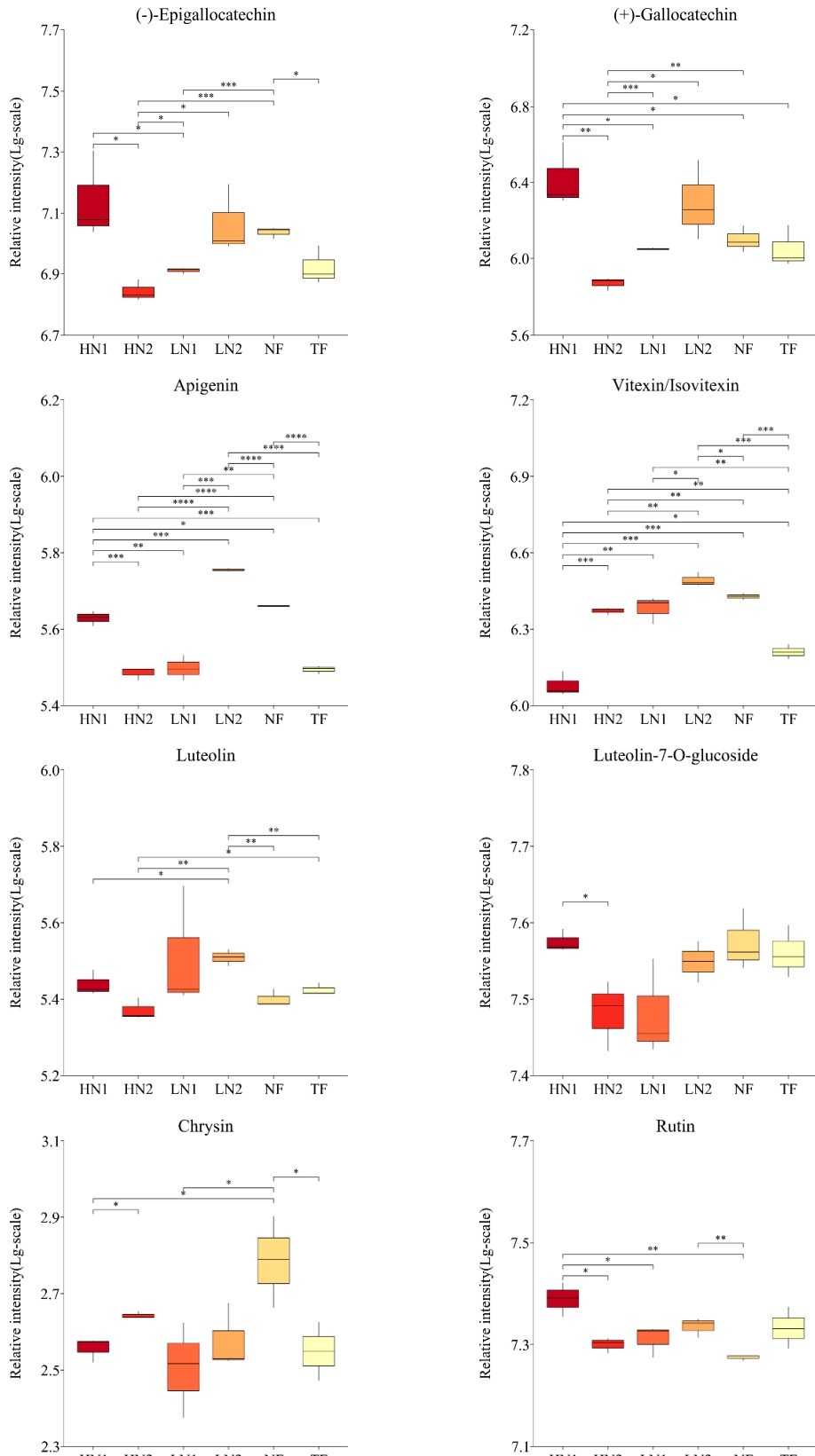

**Figure 4.** *Cont.*

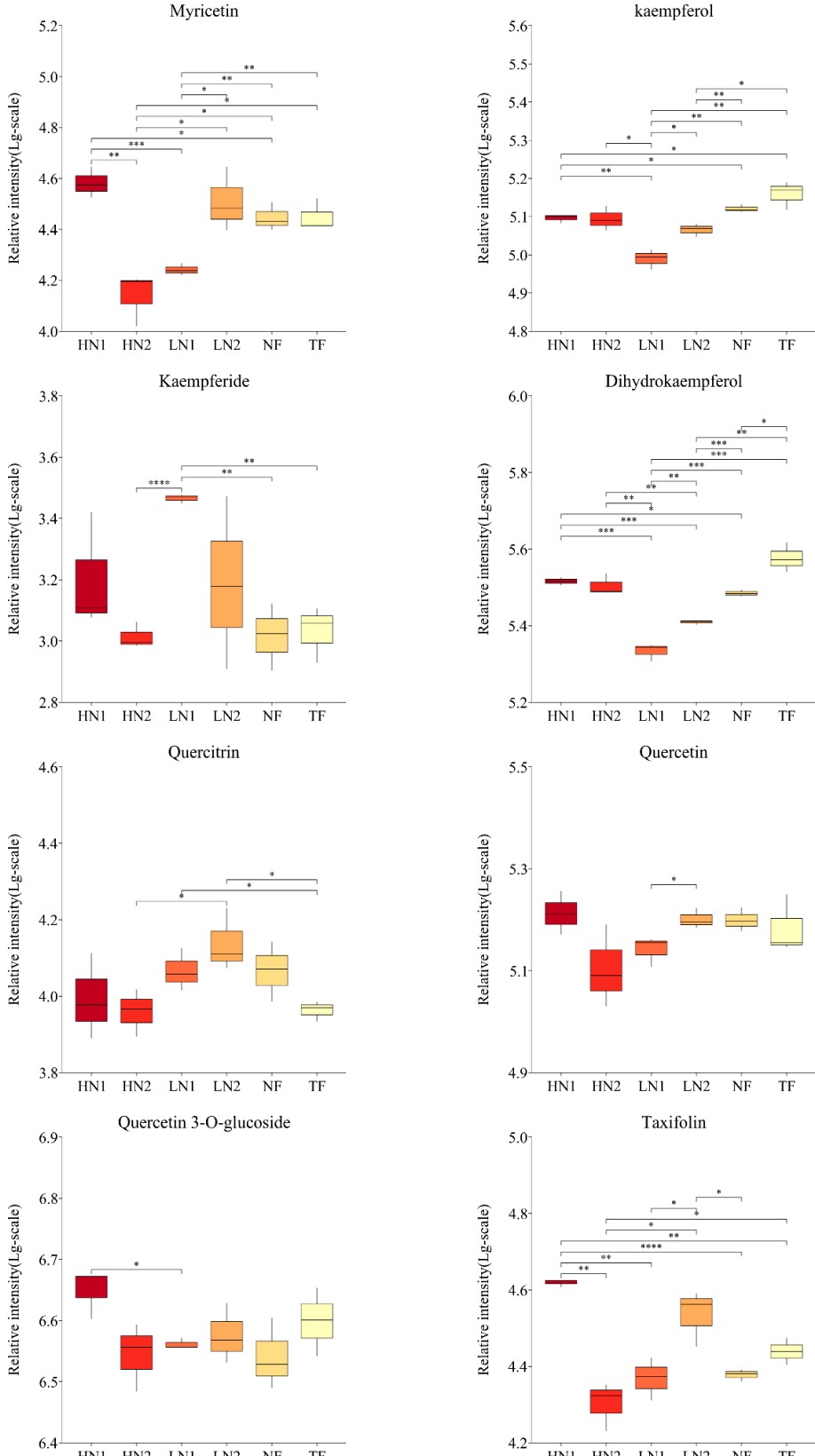

**Figure 4.** *Cont.*

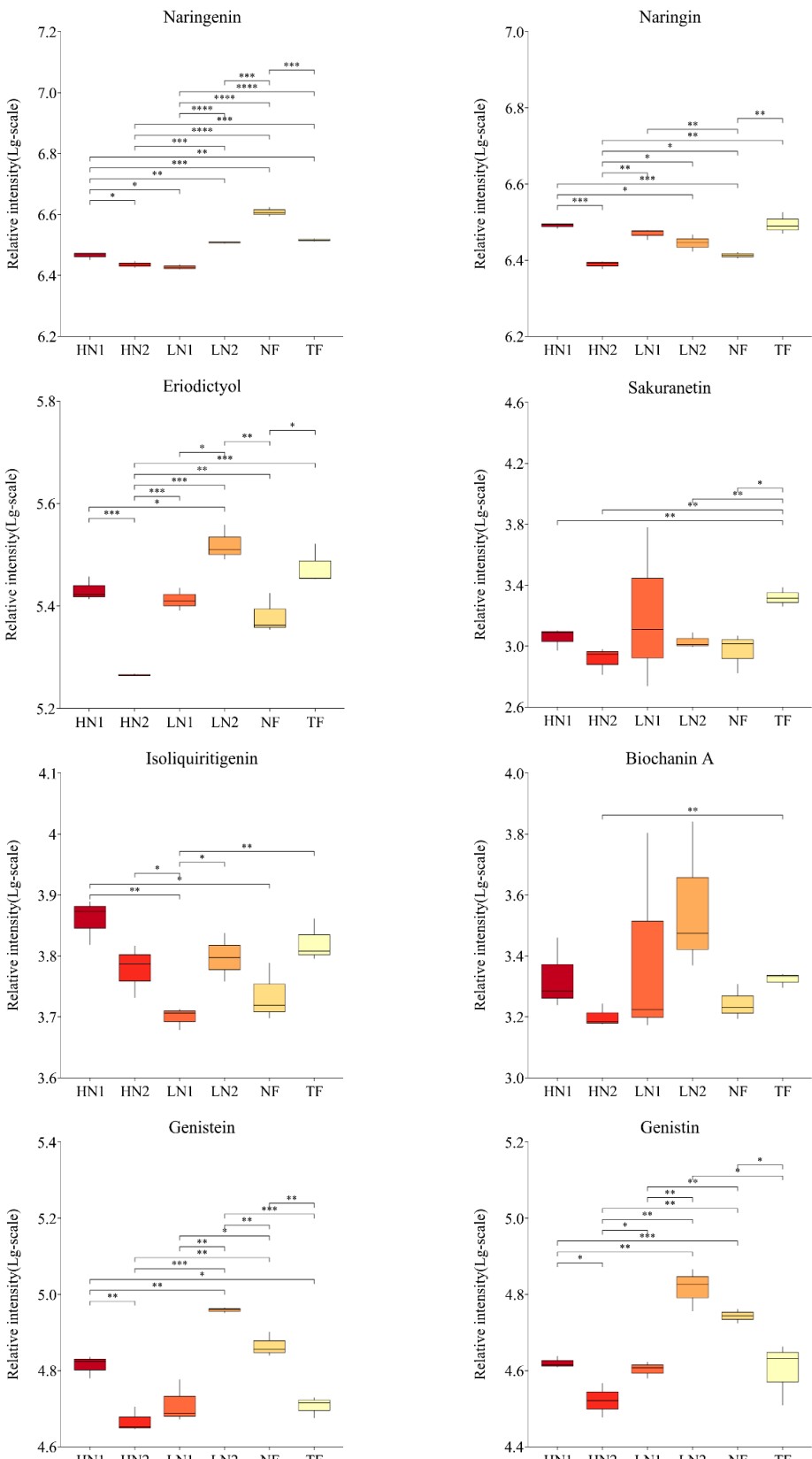

**Figure 4.** Changes in flavonoid compound accumulation in tea leaves with different N supplies. This figure shows the 26 kinds of flavonoid compounds detected. The statistical approach used was a two-tailed *t*-test. Every group contains 3 tea leaf samples. ****, ***, **, * indicates a significant difference at $p < 0.0001$, $p < 0.001$, $p < 0.01$, and $p < 0.05$, respectively.

In addition, the accumulations of 6 flavonoid ingredients (namely, catechin, epicatechin, gallocatechin, apigenin, taxifolin, and genistein) in HN1 were significantly higher than those in TF, while vitexin, kaempferol, naringenin, and sakuranetin showed higher levels in TF, indicating differences in response to the nitrogen nutrient of flavonoid metabolism between a N-starved tea plant and N-sufficient tea plant.

*3.5. Changes in Phytohormone Levels in Tea Shoots with Different N Applications*

The content changes of 15 main phytohormones in tea shoots with different N applications were analyzed (Table 2), including 4 somatotropins (IAA, GA, CKs, and BR) and 4 stress response hormones (ABA, JA, SA, and ACC). The CK content was the sum of iPR (isopentenyladenosine), iP (isopentenyladenine), transZR (trans-zeatin riboside), transZ (trans-zeatin), cisZR (cis-zeatin riboside), and cisZ (cis-zeatin) contents. Among GA1 (Gibberellin 1), GA3 (Gibberellin 3), GA4 (Gibberellin 4), and GA7 (Gibberellin 7), only GA4 was detected.

**Table 2.** The content of phytohormones in tea leaves with different N supplies.

| Plant Hormones | Content (ng/g) | | | | | |
|---|---|---|---|---|---|---|
| | **HN1** | **HN2** | **LN1** | **LN2** | **NF** | **TF** |
| ABA | 860.68 ± 44.27 c | 1181.40 ± 86.86 b | 1928.32 ± 117.73 a | 968.46 ± 15.90 c | 981.22 ± 64.17 c | 951.15 ± 8.55 c |
| cis-OPDA | 56.94 ± 9.77 b | 60.45 ± 15.45 b | 150.41 ± 35.64 a | 118.10 ± 49.28 a | 37.93 ± 3.19 b | 63.28 ± 18.77 b |
| JA | 600.13 ± 141.26 d | 2909.97 ± 344.27 a | 1669.73 ± 282.16 b | 1230.19 ± 63.84 c | 834.05 ± 45.39 d | 1475.86 ± 255.05 bc |
| JA-Ile | 102.92 ± 12.11 e | 207.98 ± 15.72 c | 146.76 ± 3.83 d | 153.35 ± 5.02 d | 245.76 ± 8.77 b | 312.28 ± 20.15 a |
| SA | 3177.61 ± 46.71 cd | 3239.47 ± 154.43 cd | 3583.45 ± 70.06 a | 3354.66 ± 91.96 bc | 3436.38 ± 104.29 ab | 3150.73 ± 110.32 d |
| ACC | 1212.41 ± 71.28 a | 873.03 ± 133.20 c | 1093.79 ± 177.18 ab | 1158.99 ± 64.34 ab | 1035.72 ± 99.41 abc | 958.49 ± 104.74 bc |
| GA4 | 2.46 ± 0.23 bc | 2.90 ± 0.57 ab | 2.02 ± 0.45 c | 2.63 ± 0.06 abc | 2.07 ± 0.28 c | 3.21 ± 0.23 a |
| IAA | 101.94 ± 16.86 b | 123.35 ± 23.57 ab | 160.65 ± 30.13 a | 112.47 ± 14.59 b | 139.31 ± 7.51 ab | 134.69 ± 30.78 ab |
| TY (Typhasterol) | 25.47 ± 1.20 a | 23.08 ± 1.51 a | 22.82 ± 1.60 a | 24.67 ± 0.83 a | 23.71 ± 1.30 a | 22.82 ± 2.53 a |
| CKs | 16.47 ± 0.25 b | 21.05 ± 1.29 a | 15.81 ± 1.09 b | 16.50 ± 3.00 b | 15.23 ± 1.95 b | 15.74 ± 1.15 b |
| iP | 0.37 ± 0.03 c | 0.52 ± 0.11 b | 0.46 ± 0.05 bc | 0.37 ± 0.06 c | 0.77 ± 0.02 a | 0.40 ± 0.03 c |
| transZ | 0.39 ± 0.08 b | 0.43 ± 0.06 b | 0.38 ± 0.01 b | 0.49 ± 0.13 b | 0.42 ± 0.06 b | 0.72 ± 0.13 a |
| cisZ | 0.82 ± 0.09 c | 1.21 ± 0.09 a | 0.84 ± 0.15 c | 0.85 ± 0.14 bc | 1.14 ± 0.12 a | 1.05 ± 0.06 ab |
| iPR | 3.56 ± 0.27 a | 3.40 ± 0.26 ab | 3.02 ± 0.27 bc | 3.02 ± 0.13 bc | 3.39 ± 0.24 ab | 2.61 ± 0.16 c |
| transZR | 5.51 ± 0.37 b | 7.95 ± 0.50 a | 4.96 ± 0.26 b | 5.52 ± 1.37 b | 4.39 ± 1.13 b | 5.11 ± 0.81 b |
| cisZR | 5.82 ± 0.43 b | 7.53 ± 0.55 a | 6.16 ± 0.70 b | 6.25 ± 1.28 ab | 5.11 ± 0.62 b | 5.84 ± 0.49 b |

Note: Values are presented by the mean ± standard error, and different lowercase letters in the same represent a significant difference; $p < 0.05$.

As shown in Table 2, the GA4 content of tea leaves that were non-fertilized (NF) was 2.07 ng/g, which was 35.67% lower than that of normal nitrogen-fertilized tea plants (TF, 3.21 ng/g), indicating that N deficiency led to the insufficient synthesis of growth-stimulating hormones. The contents of IAA, CKs, and TY have no significant changes, but the composition of the CKs showed changes, with an increase in iP and iPR content, and a decrease in transZ content. After nitrogen supplementation, the contents of CKs and GA4 both showed an upward trend. The content of CKs in HN2 was increased by 38.20% compared to NF and increased by 33.74%, compared to TF. The GA4 content of HN2 was 40.39% higher than that of NF but was not significantly different from TF. It indicated that N supply promoted the synthesis of growth-promoting hormones in tea plants, especially the massive synthesis of CKs. The IAA content of HN1 was 36.55% lower than that of LN1, indicating that a N supply in large amounts inhibited IAA synthesis. There was no significant difference in TY content among the different treatments, providing less influence for nitrogen nutrients on TY content.

For stress-responsive hormones, the SA content (3436.38 ng/g) in NF increased by 9.07%, compared to TF (3150.72 ng/g), while the JA content (834.05 ng/g) and JA-Ile (jasmonic acid-isoleucine) content (245.76 ng/g) decreased by 43.49% and 27.07%, respectively. With nitrogen supplementation, the contents of the stress-responsive hormones showed a trend of initially increasing and then decreasing with the rising nitrogen application rate. Among them, the ABA content was the highest in LN1 (1928.32 ng/g) and the JA content was the highest in HN2 (2909.97 ng/g). The ABA and SA contents in LN1, the JA, and CK contents in HN2, and the ACC contents in HN1 were 2.03, 1.14, 1.97, 1.34, and 1.26 times that in the TF, respectively.

*3.6. Correlation Analyses among Amino Acids, Flavonoids, and Phytohormones*

The correlation network analysis results of the detected amino acid, flavonoid, and phytohormone components shown in Figure 5 demonstrate that cisZ, cis-OPDA(cis-12-oxo-phytodienoic acid), JA-Ile, iP, iPR, etc., were significantly correlated with the accumulation of various amino acids or flavonoids. Specifically, cisZ was positively correlated with taxifolin, myricetin, apigenin, genistein, catechin, aspartate, and hydroxyproline, and negatively correlated with ferulic acid, p-coumaric acid, phenylalanine, and orithine. cis-OPDA was positively correlated with ferulic acid, p-coumaric acid, phenylalanine, and taurine, and negatively correlated with 8 flavonoid compounds (naringenin, luteolin-7-O-glucoside, myricetin, quercetin, apigenin, genistein, genistin, and eriodictyol), and glutamate. JA-Ile was positively correlated with genistein, genistin, naringenin, apigenin, and hydroxyproline, and negatively correlated with kaempferide. JA was positively correlated with epicatechin and catechin. iP was significantly positively correlated with apigenin, genistein, taxifolin, eriodictyol, aspartate, arginine, putrescine, and hydroxyproline, and negatively correlated with tyrosine and phenylalanine. iPR was positively correlated with aspartate, glutamate, rutin, and isoliquiritigenin, and negatively correlated with alanine and creatine. transZR was positively correlated with catechin, epicatechin, Taxifolin, and rutin, and negatively correlated with vitexin. ABA was positively correlated with taurine and glycine, and negatively correlated with genistin, genistein, eriodictyol, and glutamate. IAA was positively correlated with citrulline and taurine, and negatively correlated with naringin. SA was positively correlated with glutamine, ornithine, and p-coumaric acid, and negatively correlated with epicatechin and cystine (see Tables S4 and S5 in the Supplementary Materials for details).

Many pairs with significant correlations were observed between amino acids and flavonoid components. Chiefly, glutamate, aspartate, and hydroxyproline showed a significantly positive correlation with 8 kinds of flavonoid compounds (namely, catechin, epicatechin, gallocatechin, naringin, taxifolin, eriodictyol, myricetin, and rutin), 11 kinds of flavonoid compounds (namely, catechin, epicatechin, epigallocatechin, gallocatechin, naringin, taxifolin, eriodictyol, rutin, myricetin, isoliquiritigenin, and quercetin 3-O-glucoside), and 8 kinds of flavonoids compounds (namely, epicatechin, epigallocatechin, gallocatechin, myricetin, apigenin, genistein, genistin, and taxifolin), respectively. Phenylalanine and ornithine, which were both significantly positively correlated with ferulic acid and *p*-coumaric acid, were significantly negatively correlated with 9 and 7 kinds of flavonoid compounds, respectively (see Supplementary Table S6 for details).

Among the congeneric metabolites, glutamate and aspartate had a significantly positive correlation (Pearson's $r = 0.8667$, $p < 0.05$). These metabolites also had a significant negative correlation with some amino acids, such as tyrosine, glycine, leucine, phenylalanine, etc., while proline, valine, phenylalanine, and tyrosine showed positive relations with 17, 14, 11, and 12 amino acid compounds, respectively. This illustrated a relatively independent accumulation pattern of glutamate and aspartate in tea leaves. Meanwhile, several flavonoid components, including quercetin, quercetin-3-O-glucoside, taxifolin, myricetin, and rutin were positively related to C (catechin), EC (epicatechin), GC (gallocatechin), and EGC, any two of which were positively associated with each other. However, vitexin exhibited a negative correlation with 7 flavonoid constituents (namely, C, EC, Quercetin 3-O-glucoside, naringin, dihydrokaempferol, rutin, and isoliquiritigenin), perhaps indicating a relatively independent synthesis pathway of vitexin. In addition, 14 kinds of flavonoid constituents containing flavanols, flavonols, flavones, flavonoid glycosides, etc., were negatively related to ferulic acid (see Supplementary Tables S7 and S8 for details).

Among the different hormone components, several pairs of correlations were also found; for example, ABA content was significantly positively correlated with cis-OPDA and SA content, as well as JA-Ile with transZ and cisZ, and JA with cisZR and transZR (see Supplementary Table S9 for details).

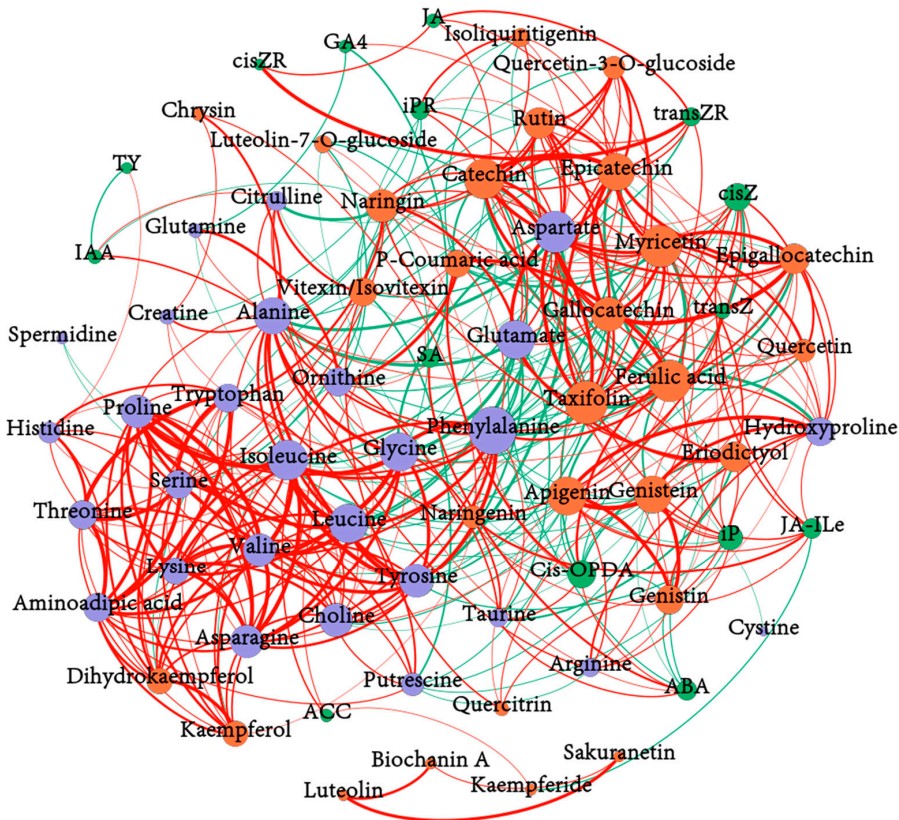

**Figure 5.** The correlation network analysis among amino acids, flavonoids, and phytohormones. $|r| > 0.5$, $p < 0.05$. The colors of the nodes represent the categories of metabolite and the size of nodes relates to the quantity of the corresponding metabolite; the red line means a positive correlation and the green line means a negative correlation. The wider the line, the more significant the correlation.

## 4. Discussion

### 4.1. Effect of Stress on Tea Plants Caused by N-Deficiency and Graded N Supply on the Regulation of Auxin Homeostasis in Relation to Amino Acids and Flavonoids

Nitrogen is essential for plant growth and development, due to its involvement in the composition of amino acids and protein enzymes, as well as many other necessary ingredients for plant cells [24]. A consistent nitrogen supply in tea gardens is beneficial to increase the soil availability of N and enrich the soil microbial community, thus promoting the growth and development of tea plants as well as tea quality. Long-term N application promotes the synthesis of chlorophyll, caffeine, and amino acids, especially L-theanine, in fresh tea leaves, but reduces that of polyphenols, aroma compounds, etc. [25]. In this study, a long-term lack of fertilization resulted in severe available nitrogen deficiency in the soil (Table 3), which was in accordance with a previous study [26]. The tea plants showed typical N-deficiency symptoms, with smaller and yellowed leaves, as well as a decline in sprouting; thus, the yield was only 42.69% of that with the TF treatment. Meanwhile, the contents of theanine and soluble sugar were reduced by 60.17% and 18.82%, respectively. These findings suggested that N-starvation and weakened carbon assimilation in tea plants are caused by soils with lower available nitrogen contents [27].

Abiotic stress, including limited nitrogen availability, generally induces the production of reactive oxygen species (ROS) in various species [28], an excess level of which ultimately results in oxidative stress and even apoptosis. Fortunately, plants have evolved an extraordinarily diverse suite of protective mechanisms against adverse conditions, mainly including non-enzymatic and enzymatic antioxidants; for instance, the up-regulated expression of antioxidant metabolites and proteolysis also alters the homeostasis of phytohormones, etc. [29,30]. In our study, the restricted growth and yellowed leaves of the tea plants, as well as the up-regulated expression of several flavonoid compounds and hydroxyproline under long-term

non-fertilization implied a degree of oxidative stress at the phenotypic and metabolic levels [29,31]. Consequently, the reduced accumulation of glutamate probably resulted from its participation in the ascorbate-glutathione cycle [32], which plays a key role in the natural antioxidant system in the plants [30]. On the other hand, an inhibited TCA cycle by oxidative stress, as well as a deficiency of available N in the soil, limited the synthesis of glutamate and its derivatives, including aspartate and theanine [33,34]. Yang et al. [33] demonstrated that glutamate and glutamate-derived amino acids are the most dynamic in response to N supply in tea plants.

**Table 3.** The physical and chemical properties of soils with NF or TF treatment.

| Treatment | Soil Layer | pH | OM g/kg | TN g/kg | TP g/kg | TK g/kg | $NO_3^-$-N mg/kg | $NH_4^+$-N mg/kg | AP mg/kg | AK mg/kg |
|---|---|---|---|---|---|---|---|---|---|---|
| NF | 0–20 cm | 4.14 | 22.0 | 1.18 | 0.43 | 13.57 | 8.33 | 3.10 | 17.60 | 63.67 |
| | 20–40 cm | 4.30 | 15.57 | 0.92 | 0.38 | 13.61 | 4.13 | 2.23 | 4.80 | 53.00 |
| TF | 0–20 cm | 3.84 | 12.6 | 1.04 | 0.46 | 8.83 | 36.85 | 76.50 | 20.30 | 138.00 |
| | 20–40 cm | 4.08 | 17.55 | 1.01 | 0.31 | 12.44 | 20.93 | 16.65 | 5.2 | 136.00 |

Note: OM, organic matter. TN, total nitrogen. TP, total phosphorus. TK, total potassium. AP, available phosphorous. AK, available potassium.

N deficiency induced the reinstitution of the plant metabolism; metabolism could be effectively restored following N resupply [35,36]. Commonly, N supply can accelerate N metabolism and increase the contents of most amino acids [2]. Interestingly, graded N supply induced an increase in the accumulation of theanine, glutamate, and aspartate, but a reduction in the accumulation of as many as 16 kinds of amino acids, such as lysine, aromatic amino acid, branched-chain amino acids, etc. Similar results have been reported by Xu et al. [25]. This may reflect the physiological mechanism of tea plant reactions during N stress release, including the enhanced metabolic pathways of the TCA cycle and GS/GOGAT cycle, as well as the usage of amino acids in low abundance as an alternative respiratory substrate, to cope with raising energy demands due to enhancing metabolism [37,38]. In addition, those amino acids with reduced accumulation were used for the biosynthesis of secondary metabolites, for instance, chlorophyll, nucleic acids, phytohormones, and other nitrogenous substances [38]. The N supply affects the accumulation of flavonoids via the regulation of C and N metabolism in tea plants, while higher N application enhanced the N metabolism, resulting in a lower accumulation of flavonoids [3,5]. However, we observed an opposite result, in that a higher N application promoted the accumulation of flavonoids rather than a lower N application during N-deficiency stress release. At the same time, ferulic acid, p-coumaric acid, and phenylalanine appeared at higher levels under a lower available N application (Figure 6), which demonstrated a weaker phenylpropanoid pathway that was perhaps caused by the downregulation of flavonoid synthesis-related enzyme [39]. Therefore, the high accumulation of flavonoids requires an optimum N supply, which is consistent with the aforesaid study conducted by Dong et al. [6]. Moreover, because of the distinction between organic N and organic N supply in terms of the available nitrogen, there were also various effects on the physicochemical properties and microflora of soil [40,41]; the response of tea plants to organic N supply obviously differed from an equal N supply in an inorganic form. In this study, for instance, there were 16 kinds of flavonoid compounds and 23 kinds of amino acids that showed a significant difference in accumulation levels between HN1 and HN2; in addition, a similar result was observed between LN1 and LN2. Furthermore, we have noted a significant increase in the accumulation of spermidine and putrescine in HN1, proving a transformation from N stress to salt stress caused by the high inorganic N application [5,37]. Furthermore, some intriguing correlations were identified by correlation network analysis. This correlativity could provide us with a better understanding of the interaction of different substances and the physiological metabolism of tea plants. Glutamate, aspartate, and phenylalanine, as well as ferulic acid, were at the center of the correlation network and played a crucial role in the regulation of N metabolism and the biosynthesis of flavonoids. In particular, glutamate

and aspartate showed a significantly positive correlation with 8 and 11 kinds of flavonoid compounds, respectively, which indicated their central role in C/N metabolism [25,42]. Besides this, the coordinated variation of flavonoids and hydroxyproline suggested the key role of flavonoids in enhancing plant tolerance to stress [31].

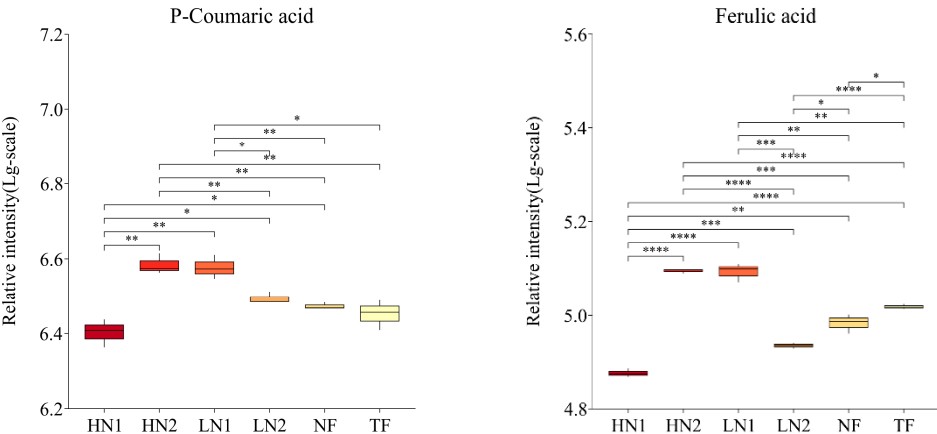

**Figure 6.** The changes in ferulic acid and p-coumaric acid contents in tea leaves with different N supplies. The statistical approach used was a two-tailed *t*-test. Every group contains 3 tea leaf samples. ****, ***, **, * indicates a significant difference at $p < 0.0001$, $p < 0.001$, $p < 0.01$, and $p < 0.05$, respectively.

Indole-3-acetic acid (IAA), the predominant auxin, is one of the most important phytohormones that regulate plant growth and development but is seldom involved in the regulation of oxidative stress factors [43]. However, much less is known about the ways that IAA is catabolized [44]. Previous research has shown that N application can regulate the endogenous auxin level, in order to achieve better plant growth [45]; meanwhile, the plant can increase its auxin concentration against nutrient stress [46]. In this study, the contents of IAA in tea leaves with continuous non-fertilization (NF) stayed at the same level as that of normal fertilization (TF), which may be related to the conjugation and hydrolysis of the auxin conjugates in the tea plant. Another possibility suggested by our data is that the defense mechanism of tea plants activated the shikimate and related pathways to promote the synthesis of indolic compounds [29]. Therefore, an older tea plant is capable of maintaining its auxin level under bearable N-deficiency stress, owing to its abundant nutrient storage in mature leaves, branches, and the trunk [47]. Tea leaves are rich in flavonoids, which have the ability to modulate auxin movement and promote auxin degradation [31]. In this study, IAA content showed a negative correlation with the accumulation of naringin. Due to the increase in the accumulation of 12 kinds of flavonoid compounds, the IAA content of tea leaves with HN1 treatment decreased markedly, compared to that of LN1.

*4.2. Effect of Stress on Tea Plants Caused by N-Deficiency and Graded N Supply on the Regulation of Homeostasis of CKs and ABA, in Relation to Amino Acids and Flavonoids*

Cytokinins (CKs), adenine derivatives, and nitrogenous substances have attracted the significant attention of more than one researcher because of their potential to improve crop yield. Those with an isoprenoid side chain are the predominant forms ofCKs, including trans-zeatin (transZ), isopentenyladenine (iP), cis-zeatin (cisZ), and dihydrozeatin, being derived from nucleotides directly and being abundant in most plants [48]. In the plant cell, CK homeostasis is affected by the combined action of isopentenyltransferase and the CKs oxidase and dehydrogenase, depending on the plant's developmental stage and growing conditions. In this study, the CK levels of tea leaves were maintained but changed in composition, with an increase in iP and iPR content and a decrease in transZ content under nutrient stress. We suppose that stress intensified the degradation of specific isoprenylated tRNA species, then the iP content, as well as the iPR content, increased [49]; nonetheless, this finding warrants further examination. Although the biological significance of the

different variants is less well-known, our results may reveal that the reduction in the conversion of iP to zeatin, and the decrease in the long-distance transportation of zeatin from roots to shoots, are important mechanisms against N stress in tea plants [50]. On the other hand, transZ-type cytokinins are critical for maximal leaf growth, which could be partially the reason for a lower yield of NF treatment. With N supply, Z-type cytokinins and iP showed completely opposite changes in tea leaves; the Z-type (transZR, cisZR) increased, while the iP decreased, suggesting an enhanced conversion of iP to the zeatin forms and the transportation of CKs during the N stress recovery period, contributing to an increase in yield. Moreover, the homeostasis of CKs is closely linked to flavonoids and amino acids; specifically, the iP forms and Z-type were significantly positively correlated with 6 and 11 kinds of flavonoids and were significantly correlated with 9 and 4 kinds of amino acids, respectively. These findings implied that the regulation of the homeostasis of CKs effected by the N supply is related to amino acids and flavonoids in tea plants.

Plant hormones and transcription factors (TFs) play a key role in the regulation of the root N absorption system, specifically, abscisic acid (ABA)-related TFs [51,52]. Ammonium and nitrate, both as N sources and signal molecules, can lead to changes in the contents of the relevant plant hormones. Here, the ABA content in tea leaves was kept up under N stress but first rose and then descended with the rising N supply, and reached its maximum level under LN1 treatment, accompanied by a lower level of flavonoids. Previous work with exogenous ABA treatment has demonstrated that ABA can activate PAL, CHS, and ANS, promoting the phenylpropanoid/flavonoid and anthocyanin pathways [53]. However, in our results, the ABA level was negatively correlated with several flavonoids (e.g., genistin, genistein, and eriodictyol), reflecting the possible feedback regulation of flavonoids on ABA.

*4.3. Effect of Stress on Tea Plants, Caused by N Deficiency, and Increased N Supply on the Regulation of Homeostasis, Mainly SA, OPDA, JA, and JA-Ile, in Relation to Amino Acids and Flavonoids*

Defense pathways mediated by jasmonic acid (JA) and salicylic acid (SA) form the backbone of the mechanisms of plant stress resistance [54]. However, the interaction of these two pathways is mainly considered to be antagonistic [55]. In this study, accordingly, there was a significant reduction in the contents of JA and JA-Ile, with an increase in the content of SA in tea leaves under nutrient stress. We propose that the degradation of unsaturated fatty acids (USFA) to oxylipins, caused by oxidative stress, was likely to be responsible for the reduction in JA and JA-Ile content. Besides this, a decrease in the quantity and volume of chloroplasts caused by the N deficiency weakened the conversion of USFA to oxo-phytodienoic acid (OPDA) [56,57]. With a N supply, a similar trend of changes that first rose and then descended occurred in cis-OPDA, JA, and SA contents, while the JA-Ile content decreased, owing to its conjugate cleavage [58], which directly proved that the homeostasis of the stress hormone in tea plants was closely related to nutrient conditions. In addition, JA and JA-Ile levels were positively correlated with several flavonoids and hydroxyproline, while the cis-OPDA level showed a significant negative correlation with 8 kinds of flavonoids and glutamate. Therefore, cis-OPDA may possibly induce the biosynthesis of glutathione from glutamate against oxidative stress, along with flavonoids to improve the conversion of cis-OPDA to JA.

## 5. Conclusions

This study considers that the response of amino acids, flavonoids, and phytohormones in tea plants to nitrogen is closely related to the nitrogen nutrient status of tea plants. Nitrogen deficiency stress induced a decrease in the accumulation of glutamate, aspartate, and theanine, but an increase in the accumulation of some flavonoid components. With N resupply, a small amount of nitrogen promoted the biosynthesis of glutamate, aspartate, and theanine, whereas it inhibited the accumulation of flavonoids and most amino acids, mainly those containing lysine, aromatic amino acids, and branched-chain amino acids while raising the nitrogen application also enhanced the synthesis of flavonoids. During

N deprivation then resupply, glutamate and aspartate, as well as hydroxyproline, played a crucial role in the metabolic adaptation of tea plants. Conversely, hormone synthesis in nitrogen-starved tea plants was more responsive to nitrogen than in nitrogen-sufficient tea plants. These results revealed the characteristics of the N utilization of tea plants during N deficiency stress release, to some degree.

**Supplementary Materials:** The following supporting information can be downloaded at: https://www.mdpi.com/article/10.3390/f13101629/s1, Table S1: UPLC-MS/MS parameters for amino acid determination; Table S2: UPLC-MS/MS parameters for flavonoid determination; Table S3: UPLC-MS/MS parameters for phytohormone determination; Table S4: The correlation between phytohormones and amino acids; Table S5: The correlations between phytohormones and phenolic compounds; Table S6: The correlations between phenolic compounds and amino acids; Table S7: The correlations among components of amino acids; Table S8: The correlations among components of phenolic compounds; Table S9: The correlations among components of phytohormones.

**Author Contributions:** Data curation, L.L., L.Q., Y.H. and M.W.; formal analysis, Y.Y.; funding acquisition, F.L.; investigation, L.L., L.Q., Y.Y. and Y.W.; methodology, F.L.; resources, Y.H.; software, Y.X.; writing—original draft, X.G.; writing—review and editing, X.G. and H.M. All authors have read and agreed to the published version of the manuscript.

**Funding:** This research was funded by the Promotion Project of Modern Agricultural Discipline Construction of Sichuan Academy of Agricultural Sciences (2021XKJS081), the National Key Research and Development Program of China (2019YFC0840500). And the APC was funded by the China Agriculture Research System of MOF and MARA (CARS-19).

**Data Availability Statement:** Not applicable.

**Acknowledgments:** We very much appreciate the support from Tea Research Institute, Sichuan Academy of Agricultural Sciences, and Institute of Quality Standard and Testing Technology Research, Sichuan Academy of Agricultural Sciences, and Nanjing Agricultural University for this research.

**Conflicts of Interest:** The authors declare that they have no conflict of interest.

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
